# Electrochemical Biosensor for the Determination of Specific Antibodies against SARS-CoV-2 Spike Protein

**DOI:** 10.3390/ijms24010718

**Published:** 2022-12-31

**Authors:** Sarunas Zukauskas, Alma Rucinskiene, Vilma Ratautaite, Almira Ramanaviciene, Greta Pilvenyte, Mikhael Bechelany, Arunas Ramanavicius

**Affiliations:** 1Department of Physical Chemistry, Institute of Chemistry, Faculty of Chemistry and Geosciences, Vilnius University (VU), Naugarduko Str. 24, LT-03225 Vilnius, Lithuania; 2Department of Nanotechnology, State Research Institute Center for Physical Sciences and Technology (FTMC), Sauletekio Av. 3, LT-10257 Vilnius, Lithuania; 3Department of Electrochemical Material Science, State Research Institute Center for Physical Sciences and Technology (FTMC), Sauletekio Av. 3, LT-10257 Vilnius, Lithuania; 4NanoTechnas—Center of Nanotechnology and Materials Science, Institute of Chemistry, Faculty of Chemistry and Geosciences, Vilnius University (VU), Naugarduko Str. 24, LT-03225 Vilnius, Lithuania; 5Institut Européen des Membranes, IEM, UMR 5635, University of Montpellier, ENSCM, CNRS, 34090 Montpellier, France

**Keywords:** coronavirus disease 2019 (COVID-19), SARS-CoV-2 virus proteins, specific antibodies, immune complex, electrochemical biosensor, immunosensor, self-assembled monolayer (SAM), polycrystalline gold electrode, cyclic voltammetry, differential pulse voltammetry, potential pulsed amperometry

## Abstract

In this article, we report the development of an electrochemical biosensor for the determination of the SARS-CoV-2 spike protein (rS). A gold disc electrode was electrochemically modified to form the nanocrystalline gold structure on the surface. Then, it was further altered by a self-assembling monolayer based on a mixture of two alkane thiols: 11-mercaptoundecanoic acid (11-MUA) and 6-mercapto-1-hexanol (6-MCOH) (SAM_mix_). After activating carboxyl groups using a N-(3-dimethylaminopropyl)-N’-ethyl-carbodiimide hydrochloride and N-hydroxysuccinimide mixture, the rS protein was covalently immobilized on the top of the SAM_mix_. This electrode was used to design an electrochemical sensor suitable for determining antibodies against the SARS-CoV-2 rS protein (anti-rS). We assessed the association between the immobilized rS protein and the anti-rS antibody present in the blood serum of a SARS-CoV-2 infected person using three electrochemical methods: cyclic voltammetry, differential pulse voltammetry, and potential pulsed amperometry. The results demonstrated that differential pulse voltammetry and potential pulsed amperometry measurements displayed similar sensitivity. In contrast, the measurements performed by cyclic voltammetry suggest that this method is the most sensitive out of the three methods applied in this research.

## 1. Introduction

The severe acute respiratory syndrome coronavirus 2 (SARS-CoV-2) virus is still a global problem, producing a burden in various countries. Therefore, the development of accurate biosensors for determining these virus proteins and/or antibodies against these proteins are essential strategies to stop or to reduce the spread of this virus.

The development of any electrochemical sensor first starts with the selection of the electrode material. Different types are commonly used to prepare electrochemical sensors, such as: gold [1,2,3], platinum [4,5], indium-doped tin oxide [6], nanocrystalline diamond [3,7,8,9], glassy carbon [3,10], graphite [11,12], and some other materials [13,14,15,16,17]. Electrochemical biosensors are one of the methods for testing SARS-CoV-2 among other methods such as enzyme-linked immunosorbent assay (ELISA) [18] and polymerase chain reaction [19].

In many types of research, gold electrodes have been used as working electrodes. Previously, it was demonstrated that the electrochemical treatment significantly changes the gold electrode surface, e.g., the voltammetric pre-treatment of a gold electrode in sulfuric acid leads to changes in the surface roughness [20]. It was demonstrated that by selecting a suitable potential window, specific gold-based nanostructures were formed on the surface of the gold electrode. Gold electrodes were pretreated by mechanical polishing with alumina slurry, chemical oxidation (e.g., with H_2_SO_4_/H_2_O_2_), or were electrochemically ‘polished’ [21]. The additives in the electrolyte used during the pre-treatment of gold yielded interesting results. Hence, the formation and characteristics of gold nanoparticles are tuned during the electrochemical pre-treatment by selecting the electrolyte composition, e.g., surfactants—tetra-dodecyl ammonium bromide [22], homopolymers used to prevent agglomeration—polyvinylpyrrolidone [23], or pyrrole to produce polypyrrole (Ppy)-coated gold nanocomposites [24].

In addition to the electromechanical treatment, there are other ways to change the surface properties. The chosen electrode materials need to be bio-compatible and easily functionalized; a commonly used example that fits these criteria could be chitosan [25]. A notable example would be creating a multi-layer chitosan/alginate structure utilizing different pH values during the formation process [26]. SAMs are used to modify the electrode surface using fetal bovine serum on a H/O-terminated diamond [27] or human immunoglobulin G (IgG) [28]. SAMs based on alkanethiols and dialkanethiols formed on gold electrodes are used in the design of various sensing systems [29,30]. Alkanethiols chemically absorb on gold; however, the details of the bonding between the sulfur head group and gold is still a subject for debate [31]. It was observed that among different types of SAMs, the ones with terminal carboxyl and methyl groups were the most reliable for establishing highly effective links with the spike protein of the SARS-CoV-2 virus [32,33].

This article focuses on designing an electrochemical biosensor to determine the concentration of antibodies against the SARS-CoV-2 spike protein (anti-rS). The study paid significant attention to the preparation of the electrode surface. Several electrochemical methods, cyclic voltammetry (CV), potential pulsed amperometry (PPA), and differential pulse voltammetry (DPV), have been applied to determine the interaction between the immobilized rS proteins and the anti-rS, which were present in the sample. The limits of detection (LOD) and quantification (LOQ) obtained from the CV, PPA, and DPV experimental results were compared. The designed electrochemical biosensor displayed the highest sensitivity towards anti-rS antibodies when the electrochemical signal was determined by cyclic voltammetry measurements.

## 2. Results and Discussion

### 2.1. Cleaning the Gold Electrode

The polycrystalline structure of the gold electrode was formed before each set of experiments, which we identified according to the representative peaks in voltammograms obtained by the CV method (Figure 1). For this purpose, the potential was cycled from −0.1 V to 1.5 V at a scan rate of 0.1 V/s in 1 M H_2_SO_4_ solution until the system stabilized. The smooth voltammogram with a cathodic peak at ~900 mV is characteristic of the bare gold electrode (Figure 1, dashed line, red). The 1 M H_2_SO_4_ solution was changed as necessary between steps. Afterwards, the electrode was electrochemically polished by creating a 2 V potential for 3 min in 1 M H_2_SO_4_ using the chronoamperometry technique. The representative peaks of the polycrystalline structure of the gold on the voltammogram were identified (Figure 1, smooth line, black). The change in the electrode due to the electrochemical treatment can be seen from the change in the voltammograms before (Figure 1, dashed line, red) and after (Figure 1, smooth line, black) electrochemical polarization. We observed a significantly increased cathodic peak at ~900 mV and the formation of new anodic peaks at 1180 mV (1st peak), 1280 mV (2nd peak), and 1390 mV (3rd peak) (Figure 1, smooth line, black).

The registered cyclic voltammograms were compared with the ones reported in previous studies at similar conditions. Some studies used a prolonged potential cycling of a polished gold electrode in 0.1 M of H_2_SO_4_ at a slow scan rate [20]. The observations of this study demonstrate that potential cycling led to changes in the anodic peak, which split into three peaks after potential cycling. Following the conclusions of the previous study and the emergence of characteristic peaks in this study (Figure 1, smooth lines, black) at 1180 mV, 1280 mV, and 1390 mV in voltammograms, it was concluded that we successfully obtained the polycrystalline structure of the gold electrode. According to the study of Sukeri and Bertotti [34], the newly formed peaks observed on cyclic voltammograms correspond to crystal planes Au 100 (Figure 1, Peak 1), Au 110 (Figure 1, Peak 2), and Au 111 (Figure 1, Peak 3).

The quality of the treatment of the electrode was assessed using the cyclic voltammetry method. The electrode’s surface was cleaned and electrochemically polished to a satisfactory degree when the distance between the oxidation and reduction peaks was ∆E < 0.1 V, as demonstrated in Figure 2. Next, the SAM_mix_ on the gold was formed according to the procedure described in the experimental setup and in past studies [1]. According to our experience accumulated in previous studies, it was known that the 11-MUA forms a stable and compact SAM film and 6-MCOH in the SAM_mix_ to facilitate access of the redox mediator to the electrode surface [1]. In the previous study, CV and electrochemical impedance spectroscopy (EIS) were applied for the evaluation of the electrode surface with SAM_mix_. This effect was not observed in a different study that only used an 11-MUA-based SAM on the gold electrode [2] or an L-Cysteine-based SAM on a screen-printed carbon electrode with gold nanostructures [16]. Among different types of SAMs, the ones with terminal carboxyl and methyl groups have been proven to be the most reliable for specifically linking with the SARS-CoV-2 spike protein [32,33]. The characteristic feature of the SAM_mix_ layer is the decreased peak height of the redox probe observed on the cyclic voltammogram or the increased values of the electron transfer resistance R_ct_, which were determined by EIS-based experiments [1]. The same study also displays what changes are expected after SAM activation using EDC/NHS coupling chemistry and the immobilization of the SARS-CoV-2 spike protein.

We tested a discrete electrode at different formation steps using the CV technique (Figure 3). Compared with the Au/SAM_mix_ electrode, the Au/SAM_mix_/EDC-NHS electrode displayed an increased cathodic current, which agrees with the observations reported by other authors [35]. By forming the Au/SAM_mix_/rS electrode, we observed notably reduced cathodic and anodic current densities with a further reduction after blocking the system with BSA.

### 2.2. Electrochemical Assessment of the Developed Biosensor

In the following part of the study, the electrochemical sensor, according to the protocol described in the experimental part, is constructed and used for the detection of anti-rS antibodies.

The anodic peak on the cyclic voltammogram was used as the analytical signal. Due to the peaks becoming visually indistinct at some concentrations of anti-rS, we used the second differential to locate their position (Figure 4A). J_0_ represents the reference current density value; it is a signal measured with the sensor after blocking it by BSA and in the absence of specific antibodies (C = 0 nM). J_x_ is the measurement where the x-axis corresponds to the concentration of anti-rS antibodies. The data response (∆J = J_x_ − J_0_) follows an exponential curve; as such, we used the first measurement point, which is still linear, to determine the LOD and LOQ.

During the evaluation of the affinity interaction of the immobilized SARS-CoV-2 spike proteins with the anti-rS antibodies on the polycrystalline gold electrode by DPV, analogously to the peaks on cyclic voltammograms, the peak maximum on the voltammogram was used as the analytical signal (Figure 4B). The amplitude between the anodic current maximum and the cathodic current minimum averaged out over 10 pulses was interpreted as the analytical signal of PPA (Figure 4C). It was found that the linear range of the ∆J signal was up to 6 nM concentration of anti-rS antibodies in CV and DPV measurements and up to 12 nM concentration of anti-rS antibodies in PPA measurements. The LOD and LOQ values of three different electrochemical methods are summarized in Table 1.

DPV and PPA measurements produced similar values for the LOD and LOQ, while the measurements obtained using CV produced values nearly half of those, suggesting higher sensitivity. However, CV generated a significantly higher systematic deviation error. The comparison of LOD and LOQ values obtained by the three electrochemical methods demonstrated that the difference between the analytical signals determined by the different methods is insignificant. Therefore, for further experiments, only DPV was applied.

To assess the specificity of the system, we tested the reaction of the Au/SAM_mix_/rS electrode to the incubation in a blood serum sample of a person who was not infected by SARS-CoV-2 and was not vaccinated against COVID-19 (Figure 5, black line) and compared the results to the interaction observed when tested with the blood serum of an individual that was infected with SARS-CoV-2 and vaccinated against COVID-19 (Figure 5, red dashed line). Throughout the test, we observed all the characteristic peaks at expected positions. We tested two discrete Au/SAM_mix_/rS electrodes blocked with BSA. After the blocking the system with BSA we calculated their current density values and used them as our base value, which we subtracted from the values determined during further measurements to calculate the biosensor response. We then tested the respective electrode interaction with two sets of 1 nM and 60 nM blood serum samples, one with anti-rS antibodies and the other without anti-rS antibodies. We observed current density changes in both systems. The system tested with blood serum without anti-rS antibodies expressed a current density delta of ~20 µA/cm^2^; an equivalent system tested with anti-rS antibodies displayed a change in current density of ~160 µA/cm^2^, around eight times larger. From testing, we can observe some nonspecific bonding in the system; however, the response was significantly lower than that determined after the interaction with the serum sample of the person who was not infected by SARS-CoV-2 and was not vaccinated against COVID-19.

A summary of different electrochemical sensors for the detection of SARS-CoV-2 is demonstrated in Table 2.

## 3. Experimental Setup

### 3.1. Reagents and Instrumentation

Reagents: the H_2_SO_4_ (96%, CAS# 7664-93-9), phosphate-buffered saline (PBS) tablets (pH 7.4), ethanol (EtOH) (99.9%, CAS# 64-17-5), 6-mercapto-1-hexanol (6-MCOH) (97%, CAS# 1633-78-9), 11-mercaptoundecanoic acid (11-MUA) (98%, CAS# 71310-21-9), N-(3-dimethylaminopropyl)-N’-ethyl-carbodiimide hydrochloride (EDC) (≥99.0%, CAS# 25952-53-8), K_3_[Fe(CN)_6_] (≥99.0%, CAS# 13746-66-2), K_4_[Fe(CN)_6_] (≥99.0%, CAS# 14459-95-1), and bovine serum albumin (BSA) (>98.0%, CAS# 90604-29-8) were purchased from Sigma–Aldrich (Steinheim, Germany). The N-hydroxysuccinimide (NHS) (98.0%, CAS# 6066-82-6) was purchased from Alfa Aesar (Karlsruhe, Germany).

The SARS-CoV-2 rS proteins were prepared by UAB Baltymas (Vilnius, Lithuania) according to the procedure previously described by Liustrovaite et al. [1].

### 3.2. Serum Sample Collection and Preparation of the Analysis

Anti-rS antibodies were prepared according to the protocol reported in our previous research [1]. Serum samples were collected from a patient who, four weeks before taking the blood sample, was vaccinated by a single dose of the Vaxzevria (initially named AstraZeneca) COVID-19 vaccine two weeks before being infected by the SARS-CoV-2 virus. A positive polymerase chain reaction test confirmed the infection, and the patient displayed apparent COVID-19 symptoms. Whole-blood samples of this patient were collected into Vacuette^®^ tubes with 3.5 mL of CAT Serum Separator Clot Activator (Greiner Bio-One GmbH, Kremsmünster, Austria). The serum was separated from the whole-blood sample by centrifugation (5000 rpm with a radius of 9.5 cm/RCF 2655 g for 15 min). The concentration of the specific anti-rS antibodies in the serum was determined by a chemiluminescent microparticle-based immunoassay, and it was measured to be 4666 BAU/mL. The anti-rS concentration was converted from BAU/mL to nM by using the molecular weight of the anti-rS antibodies; the molecular weight of the IgG-class anti-rS antibodies was 150 kDa, and it was recalculated to the ratio of 1 BAU/mL:20 ng/mL [44,45,46]. The collected serum was stored at −20 °C. All biological samples were handled and collected according to the regulations determined by the Lithuanian law of ethics and were officially confirmed by the Vilnius Regional Biomedical Research Ethics Committee.

Milli-Q water (conductivity 0.055 µS/cm) was used for cleaning the gold disc electrode surface and for preparation of the phosphate-buffered saline (PBS) solutions. All chemicals were of analytical reagent grade and were used as received from the producers unless otherwise stated. All electrochemical measurements were performed in a 0.1 M PBS solution, pH 7.4, with 2 mM of [Fe(CN)_6_]^3−^ and 2 mM of [Fe(CN)_6_]^4−^.

### 3.3. Instrumentation

The µAUTOLAB TYPE III potentiostat from Metrohm (Utrecht, The Netherlands) was controlled by Nova 2.1 software (Utrecht, The Netherlands) and was used for electrochemical measurements. The measurements were performed in a three-electrode cell. The gold disc (the surface area was 0.071 cm^2^) was used as the working electrode, Pt was the counter electrode, and Ag/AgCl_(3M KCl)_ from ItalSens (Houten, The Netherlands) was the reference electrode.

### 3.4. Cleaning and Assessing of the Gold Disc Electrode

The polycrystalline structure on the surface of the gold disc electrode was formed before each set of experiments. Firstly, the surface of the gold electrode was polished using 0.3 µm of alumina slurry and cleaned in Milli-Q water for 20 min using sonification. The polycrystalline structure was formed using electrochemical polarization techniques in a two-stage process: (1) the potential was cycled from −0.1 V to 1.5 V at a potential scan rate of 0.1 V/s in a 1 M H_2_SO_4_ solution by CV until the system stabilized; (2) the electrode surface was electrochemically polished by chronoamperometry at a potential of 2 V in a 1 M H_2_SO_4_ solution for 3 min.

The quality of the electrode pre-treatment was assessed by measuring the potential difference between the redox probe’s oxidation and reduction peaks in cyclic voltammograms. We used 2 mM of [Fe(CN)_6_]^3−^ and 2 mM of [Fe(CN)_6_]^4−^ (ferrocyanide and ferricyanide) in a PBS solution (pH 7.4) as the redox probe.

### 3.5. Protein Immobilization

A mixed self-assembling monolayer (SAM_mix_) was obtained by immersing the pretreated gold electrode in a 1 mL ethanol solution with alkanethiols for 16 h. The alkanethiols solution was prepared by mixing 1 mM 6-MCOH and 1 mM 11-MUA solutions in ethanol in a volumetric ratio of 9 parts 6-MCOH and 1 part 11-MUA, resulting in an alkanethiols ratio of 9:1. After 16 h in the alkanethiol solution, the electrode was removed and washed with ethanol, dried in a nitrogen gas stream, and submerged in Milli-Q water for 2 h to remove any excess reagents. The EDC-NHS mixture (0.04 M EDC and 0.01 M NHS in PBS) was used to activate the carboxylic groups of SAM_mix_ that formed on the surface of the Au electrode (Au/SAM_mix_), thereby creating a new Au/SAM_mix_/EDC-NHS structure. EDC-NHS develops an intermediary active ester (Figure 1, Step 3). The Au/ SAM_mix_/EDC-NHS formation step lasted for 20 min and was performed in a dark environment protected from direct light exposure. Afterwards, the Au/SAM_mix_/EDC-NHS-modified electrode was exposed to the SARS-CoV-2 spike protein. Au/SAM_mix_/rS structure was formed by immersing the Au/SAM_mix_/EDC-NHS-modified electrode into a 50 µg/mL rS protein solution in PBS at room temperature for 30 min. The rS protein covalently couples via primary amine functional groups (Figure 1, Step 4). The remaining active esters are blocked with BSA by immersing the Au/SAM_mix_/rS electrode in 0.5% BSA for 30 min (Figure 1, Step 5).

### 3.6. Assessment of SAM_mix_ Formation, Protein Immobilization, and Biosensor Response towards Anti-rS Antibodies by Cyclic Voltammetry, Differential Pulse Voltammetry, and Potential Pulsed Amperometry Methods

Au/SAM_mix_/rS electrodes formed according to the protocol described in the ‘Experimental part’ and were used to determine specific anti-rS antibodies. The sensitivity of Au/SAM_mix_/rS towards anti-rS was performed in a 100 µL electrochemical cell filled with PBS and serum with anti-rS antibodies, which were added in corresponding amounts to form anti-rS concentrations in a range between 1 and 95 nM. The determination of anti-rS with Au/SAM_mix_/rS electrodes was performed by incubating Au/SAM_mix_/rS electrode at room temperature for 40 min in the solution containing anti-rS serum, thus forming an Au/SAM_mix_/rS/anti-rS structure. The anti-rS was repeatedly determined by the incubation of Au/SAM_mix_/rS electrodes in solutions with different anti-rS concentrations. The electrodes after the formation of immune complexes were thoroughly rinsed in a PBS solution, and then electrochemical measurements were performed in PBS containing 2 mM of [Fe(CN)_6_]^3−^ and 2 mM of [Fe(CN)_6_]^4−^ as a redox probe.

Three electrochemical methods were used to evaluate the immobilization of rS proteins on the electrode surface and to provide an assessment of the interaction of the Au/SAM_mix_/rS electrode with the anti-rS antibodies, namely: the CV, DPV, and PPA methods. Electrochemical measurements were performed in 0.1 M PBS, pH 7.4, with 2 mM of [Fe(CN)_6_]^3−^ and 2 mM of [Fe(CN)_6_]^4−^ as a redox probe. The potential was swept from –0.1 V to +0.6 V vs. Ag/AgCl_(3M KCl)_ at a scan rate of 0.05 V/s for the 2 CV cycles. For DPV, the potential was swept from –0.1 V to +0.6 V vs. Ag/AgCl_(3M KCl)_ with a pulse height of 25 mV, a pulse width of 50 ms, a pulse period of 100 ms, and step potential of 5 mV. For PPA, the samples were analyzed using a sequence of 10 potential pulses of +0.6 V vs. Ag/AgCl_(3M KCl)_ lasting for 2 s; between these pulses, 0 V vs. Ag/AgCl_(3M KCl)_ was applied for 2 s.

The limit of detection (LOD) and limit of quantification (LOQ) were calculated according to Equations (1) and (2):LOD = 3.3 σ/S(1)
LOQ = 10 σ/S(2)
where σ is the standard deviation and S is the slope value.

The layered composition of the electrochemical biosensor based on the polycrystalline gold electrode modified with the SAM_mix_ and proteins is presented in Figure 1.

## 4. Conclusions

The appearance of a nanocrystalline structure on the gold electrode surface was proved by the emergence of characteristic anodic peaks at ~1.2 V, ~1.3 V, and ~1.4 V on the cyclic voltammogram after electrochemical treatment. These anodic peaks proved that the nanocrystalline structure on the gold electrode surface suitable for the formation of SAM was formed. The interaction of the rS and anti-rS was evaluated by three electrochemical methods: (i) CV, (ii) DPV, and (iii) PPA. A comparison of the LOD and LOQ values demonstrated that DPV and PPA measurements produced similar values of the LOD and LOQ, while the measurements obtained using CV produced values that were nearly half of what had been obtained from the other methods, which would suggest higher sensitivity. The final part of the study was dedicated to the assessment of the specificity of the system. For this purpose, the sensors’ reaction to a blood serum sample of a person who was not infected by the SARS-CoV-2 virus and was not vaccinated against COVID-19 was tested. It was found that despite the nonspecific bonding in the system, the response of the serum sample of a person that did not contract the COVID-19 disease was significantly lower than that determined after the incubation of gold electrodes modified by spike protein of SARS-CoV-2 virus in the serum sample of a person who was infected by the SARS-CoV-2 virus and was not vaccinated against COVID-19.

## Data Availability

Not applicable.

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
