# Peer review of "Electrochemical Biosensor for the Determination of Specific Antibodies against SARS-CoV-2 Spike Protein"

_ijms, 2022, doi:10.3390/ijms24010718_

Round 1

Reviewer 1 Report

The research reported an electrochemical sensor for the determination of the SARS-CoV-2 spike protein (SARS-CoV-2-rS) and three electrochemical methods were compared. The research has certain value and significance, but discussion is too simple.

1 Carefully modify the superscript and subscript, especially the chemical formula. The format is (IS-AG/AGCL.AQ.RE)?

2 The relationship of anti-rS and anti-rS antibody, how to prepare anti-rS antibody?

3 Table 1 shown the results by three methods, Was there any significant difference. Whether the activity of protein affects the binding with the antibody, or whether it affects the test results

4 Figure 4, the concentration or unitwhat‘s “the system”, which method used to detect?

Author Response

Manuscript ‘Electrochemical biosensor for the determination of specific antibodies against SARS-CoV-2 spike protein’

After recommended revisions

Response to reviewer #1:

We would like to thank the reviewer for the review of our manuscript, valuable comments and recommendations. We did our best in order to improve the manuscript according to minor revisions recommended by all three reviewers. All the most important changes are highlighted in the revised manuscript.

Reviewer #1 wrote: The research reported an electrochemical sensor for the determination of the SARS-CoV-2 spike protein (SARS-CoV-2-rS) and three electrochemical methods were compared. The research has certain value and significance, but the discussion is too simple.

Response to the reviewer: We will thank the reviewer for positive comments on Significance level and recommendations. We have advanced discussion and many other parts of the manuscript.

Reviewer #1 wrote: Carefully modify the superscript and subscript, especially the chemical formula. The format is (IS-AG/AGCL.AQ.RE)?

Response to the reviewer: Many thanks for the comments we have improved formula and other text with the superscript and subscript.

Reviewer #1 wrote: The relationship of anti-rS and anti-rS antibody, how to prepare anti-rS antibody?

Response to the reviewer: Many thanks for the comments, we have added additional Info on preparation of anti-rS antibody samples. Samples were obtained by separating the blood serum of an individual that was confirmed to have SARS-CoV-2 infection.

Reviewer #1 wrote: Table 1 shown the results by three methods, Was there any significant difference. Whether the activity of protein affects the binding with the antibody, or whether it affects the test results.

Response to the reviewer: Many thanks for the comments, we have compared results. All three electrochemical methods displayed rather similar linear response ranges. However, the results of cyclic voltammetry demonstrated, that this method has a lower limit of detection compared to the other two methods.

Reviewer #1 wrote:  the concentration or unit?what‘s “the system”, which method used to detect?

Response to the reviewer: Many thanks for the comments, we have improved this issue.

The concentration of the blood serum with the antibodies now are shown. Two different concentrations of blood serum (1nM and 60nM), were assessed using differential pulse voltammetry. The difference in the response of the sensor was compared after incubation in the blood serum of an individual who was infected by SARS-CoV-2 virus and contained anti-rS antibodies and after incubation in blood serum of individual who was not infected by SARS-CoV-2 virus. The text has been upgraded to make this issue clear.

Many thanks for the positive feedback.

We thank you for the attention you will pay to this revised version of the manuscript and we sincerely hope that our work after these revisions will be considered as relevant and attractive for publishing.

Yours sincerely,

Arunas Ramanavicius

----------------------------------------------------------------
Prof. habil. dr. Arunas Ramanavicius

Head of Department of Physical Chemistry,

Faculty of Chemistry and Geosciences, Vilnius University,  

e-mail: arunas.ramanavicius@chf.vu.lt

Reviewer 2 Report

Congratulations to the authors.

The experimental setup and results interpretation are clearly presented.

Author Response

Manuscript ‘Electrochemical biosensor for the determination of specific antibodies against SARS-CoV-2 spike protein’

After recommended revisions

Response to reviewer #2:

We would like to thank the reviewer for the review of our manuscript, valuable comments and recommendations. We did our best in order to improve the manuscript according to minor revisions recommended by all three reviewers. All the most important changes are highlighted in the revised manuscript.

Reviewer #2 wrote: Congratulations to the authors. The experimental setup and results interpretation are clearly presented.

Response to the reviewer: We will thank the reviewer for positive comments on Significance level.

Many thanks for the positive feedback.

We thank you for the attention you will pay to this revised version of the manuscript and we sincerely hope that our work after these revisions will be considered as relevant and attractive for publishing.

Yours sincerely,

Arunas Ramanavicius

----------------------------------------------------------------
Prof. habil. dr. Arunas Ramanavicius

Head of Department of Physical Chemistry,

Faculty of Chemistry and Geosciences, Vilnius University, 

e-mail: arunas.ramanavicius@chf.vu.lt

Reviewer 3 Report

This manuscript (ijms-2043768) reported an electrochemical sensor for the determination of the SARS-CoV-2 spike protein. As indeed stated by the authors, the novelty of this work lies in the form of the nanocrystalline gold structure and a self-assembling monolayer based on two alkane thiols. However, the work is not substantially different from others previously published and other journals in the biosensing field. And this work is similar to the article “Towards an Electrochemical Immunosensor for the Detection of Antibodies against SARS-CoV-2 Spike Protein” published by Journal of The Electrochemical SocietyIn my opinion, I do not recommend for this paper be published in a high-impact journal such as International Journal of Molecular Sciences.

To improve this manuscript, several issues should be addressed as follows:

1. There are many expression errors in the article. For example, in section 1, “It was demonstrated that by selecting a suitable potential window specific types of gold-based nanostructures were formed on the surface of gold electrode.” This sentence could be modified. Please modify and check the full manuscript.

2. In scheme 1, the figure legends should be added.

3. There is only one section (section 3.1. Cleaning the gold electrode) in results and discussions, please check carefully.

4. The impact of each step of the modification of the working electrode surface on the conductivity should be analyzed.

5. Please add the experimental result of the optimization of detection conditions in section 3.

6. Please add calibration curves obtained from Cyclic voltammetry, Differential pulse amperometry, and Potential pulsed amperometry. Detailed progress to calculate the detection limit and linear range should be added.

7. In Figure 4, please add more interferent to prove the specificity of the system and give detailed figure legends.

8. In Table 2, please add more references about detecting other antibodies rather than SARS protein.

9. Please check carefully the format of the reference, such as title capitalization.

Author Response

Manuscript ‘Electrochemical biosensor for the determination of specific antibodies against SARS-CoV-2 spike protein’

After recommended revisions

Response to reviewer #3:

We would like to thank the reviewer for the review of our manuscript, valuable comments and recommendations. We did our best in order to improve the manuscript according to minor revisions recommended by all three reviewers. All the most important changes are highlighted in the revised manuscript.

Reviewer #3 wrote: To improve this manuscript, several issues should be addressed as follows:

Response to the reviewer: We will thank the reviewer for comments and recommendations.

Reviewer #3 wrote: There are many expression errors in the article. For example, in section 1, “It was demonstrated that by selecting a suitable potential window specific types of gold-based nanostructures were formed on the surface of the gold electrode.” This sentence could be modified. Please modify and check the full manuscript.

Response to the reviewer: Many thanks for the comments, we have improved the whole text.

Reviewer #3 wrote: In scheme 1, the figure legends should be added.

Response to the reviewer: Many thanks for the comments, we have improved the schematic.

Reviewer #3 wrote: There is only one section (section 3.1. Cleaning the gold electrode) in results and discussions, please check carefully.

Response to the reviewer: Many thanks for the comments, we have improved the text.

Reviewer #3 wrote: The impact of each step of the modification of the working electrode surface on the conductivity should be analyzed.

Response to the reviewer: Many thanks for the comments we have added the additional figure to the text (Fig. 3) with cyclic voltammograms after each modification step. Some comments about these changes are added to the text.

Reviewer #3 wrote: Please add the experimental result of the optimization of detection conditions in section 3.

Response to the reviewer: Many thanks for the comments, we have improved both experimental part and Results and discussion.

Reviewer #3 wrote: Please add calibration curves obtained from Cyclic voltammetry, Differential pulse amperometry, and Potential pulsed amperometry. Detailed progress to calculate the detection limit and linear range should be added.

Response to the reviewer: Many thanks for the comments, we have improved this issue and added some additional results.

Reviewer #3 wrote: In Figure 4, please add more interferent to prove the specificity of the system and give detailed figure legends.

Response to the reviewer: Many thanks for the comments we have improved this issue in recent version of the manuscript. The control sample was taken from the volunteer, who was not infected by SARS-CoV-2 virus.

Reviewer #3 wrote: In Table 2, please add more references about detecting other antibodies rather than SARS protein.

Response to the reviewer: Many thanks for the comments, we have advanced the list of references and related discussion parts.

Reviewer #3 wrote: Please check carefully the format of the reference, such as title capitalization.

Response to the reviewer: Many thanks for the comments, we checked the references and fixed detected mismatches.

Many thanks for the positive feedback.

We thank you for the attention you will pay to this revised version of the manuscript and we sincerely hope that our work after these revisions will be considered as relevant and attractive for publishing.

Yours sincerely,

Arunas Ramanavicius

----------------------------------------------------------------
Prof. habil. dr. Arunas Ramanavicius

Head of Department of Physical Chemistry,

Faculty of Chemistry and Geosciences, Vilnius University, 

e-mail: arunas.ramanavicius@chf.vu.lt

Reviewer 4 Report

In this submission, the authors report the preparation, surface modification, and antigen immobilization of the polycrystalline Au electrode and apply it for detection of SARS-CoV-2 anti-rS. This sensor shows good performance which could be used for Covid detection. The topic is interesting and could attract wide readership from scientists working in the biosensor area. Therefore, I recommend its publication after the following issues are addressed.     

1. The authors describe the potential used for the preparation of polycrystalline Au, but the information of the reference electrode is not provided.

2. Is there any morphological change of the Au electrode after electrochemical treatment?

3. Δi in Figure 2 should be ΔE.

4. The description in the sentence “the resulting observed ΔJ reduction for the 301 control sample is 8 times smaller than that for the sample of person, which was not infected by SARS-CoV-2 and was not vaccinated against COVID-19” (line 300-302) should be checked.

5. In table 2, the LOD of this work is not shown.

6. Relevant representative literatures should be cited such as Langmuir 2007, 23, 13046-13052; Critical Reviews in Biotechnology, DOI: 10.1080/07388551.2022.2037503; Sensors and Actuators A 2010, 163, 42–47.  

Author Response

Manuscript ‘Electrochemical biosensor for the determination of specific antibodies against SARS-CoV-2 spike protein’

After recommended revisions

Response to reviewer #4:

We would like to thank the reviewer for the review of our manuscript, valuable comments and recommendations. We did our best in order to improve the manuscript according to minor revisions recommended by all three reviewers. All the most important changes are highlighted in the revised manuscript.

Reviewer #4 wrote: In this submission, the authors report the preparation, surface modification, and antigen immobilization of the polycrystalline Au electrode and apply it for detection of SARS-CoV-2 anti-rS. This sensor shows good performance which could be used for Covid detection. The topic is interesting and could attract wide readership from scientists working in the biosensor area. Therefore, I recommend its publication after the following issues are addressed.

Response to the reviewer: We will thank the reviewer for positive comments and recommendations.

Reviewer #4 wrote: The authors describe the potential used for the preparation of polycrystalline Au, but the information of the reference electrode is not provided.

Response to the reviewer: Many thanks for the comments we have improved this issue; now it is mentioned that silver/silver chloride electrode in 3M KCl was used as the reference electrode for all electrochemical experiments, this information now is added into corresponding part of the text.

Reviewer #4 wrote: Is there any morphological change of the Au electrode after electrochemical treatment?

Response to the reviewer: A slight visual change in colour saturation was observed after the electrochemical treatment.

Reviewer #4 wrote: Δi in Figure 2 should be ΔE.

Response to the reviewer: Many thanks for the comments we have corrected Figure 2.

Reviewer #4 wrote: The description in the sentence “the resulting observed ΔJ reduction for the 301 control sample is 8 times smaller than that for the sample of person, which was not infected by SARS-CoV-2 and was not vaccinated against COVID-19” (line 300-302) should be checked.

Response to the reviewer: Many thanks for the comments we have improved corresponding part of the manuscript.

Reviewer #4 wrote: In table 2, the LOD of this work is not shown.

Response to the reviewer: Many thanks for the comments we have advanced table has been advanced to show here mentioned part.

Reviewer #4 wrote: Relevant representative literatures should be cited such as Langmuir 2007, 23, 13046-13052; Critical Reviews in Biotechnology, DOI: 10.1080/07388551.2022.2037503; Sensors and Actuators A 2010, 163, 42–47.  

Response to the reviewer: Many thanks for the comments we have overviewed these and some other additional references recommended by reviewers.

Many thanks for the positive feedback.

We thank you for the attention you will pay to this revised version of the manuscript and we sincerely hope that our work after these revisions will be considered as relevant and attractive for publishing.

Yours sincerely,

Arunas Ramanavicius

----------------------------------------------------------------
Prof. habil. dr. Arunas Ramanavicius

Head of Department of Physical Chemistry,

Faculty of Chemistry and Geosciences, Vilnius University, 

e-mail: arunas.ramanavicius@chf.vu.lt

Reviewer 5 Report

This paper reported the development of an electrochemical sensor for the determination of the SARS-CoV-2-rS, which is very helpful and useful. This paper is well-written and clear. Some comments are as follows:

1, There are three cathodic peaks in Fig.1. Please explain why they occur.

Author Response

Manuscript ‘Electrochemical biosensor for the determination of specific antibodies against SARS-CoV-2 spike protein’

After recommended revisions

Response to reviewer #5:

We would like to thank the reviewer for the review of our manuscript, valuable comments and recommendations. We did our best in order to improve the manuscript according to minor revisions recommended by all three reviewers. All the most important changes are highlighted in the revised manuscript.

Reviewer #5 wrote: This paper reported the development of an electrochemical sensor for the determination of the SARS-CoV-2-rS, which is very helpful and useful. This paper is well-written and clear.

Response to the reviewer: We will thank the reviewer for positive comments on Significance level and recommendations.

Reviewer #5 wrote: Some comments are as follows: There are three cathodic peaks in Fig.1. Please explain why they occur.

Response to the reviewer: Many thanks for the comments we have improved: the three cathodic peaks confirm the formation of polycrystalline gold and, according to the literature, these peaks correspond to 100, 110, and 111 crystal planes of gold. We’ve adjusted the text to make this clear.

Many thanks for the positive feedback.

We thank you for the attention you will pay to this revised version of the manuscript and we sincerely hope that our work after these revisions will be considered as relevant and attractive for publishing.

Yours sincerely,

Arunas Ramanavicius

----------------------------------------------------------------
Prof. habil. dr. Arunas Ramanavicius

Head of Department of Physical Chemistry,

Faculty of Chemistry and Geosciences, Vilnius University, 

e-mail: arunas.ramanavicius@chf.vu.lt

Reviewer 6 Report

The manuscript describes an electrochemical sensor for the determination of the SARS-CoV-2 spike protein. The electrocatalytic activity of the as-prepared sensors has been studied by CV. The manuscript can be considered for publication after some major revisions. My major concerns are listed as follow:

1.     Many graphs are not clear. The colour and clarity of figures has to be improved.

2.     The stability data of the sensor should be given in the Figures.

3.     What is the equivalent circuit of the EIS data? It should be given in the Figures.

4.     The TEM or SEM images of Au/SAMmix/EDC-NHS-modified electrode should be given in the manuscript.

5.     In the section of Introduction, some related electrochemical works should be cited (such as Journal of The Electrochemical Society, 2022, 169(2): 027504; Journal of The Electrochemical Society, 2015, 162(7): B173-B179; Journal of Solid State Electrochemistry, 2014, 18(9): 2435-2442).

Author Response

Manuscript ‘Electrochemical biosensor for the determination of specific antibodies against SARS-CoV-2 spike protein’

After recommended revisions

Response to reviewer #6:

We would like to thank the reviewer for the review of our manuscript, valuable comments and recommendations. We did our best in order to improve the manuscript according to minor revisions recommended by all three reviewers. All the most important changes are highlighted in the revised manuscript.

Reviewer #6 wrote: The manuscript describes an electrochemical sensor for the determination of the SARS-CoV-2 spike protein. The electrocatalytic activity of the as-prepared sensors has been studied by CV. The manuscript can be considered for publication after some major revisions.

Response to the reviewer: We will thank the reviewer for positive comments on Significance level and recommendations.

Reviewer #6 wrote: My major concerns are listed as follow: Many graphs are not clear. The colour and clarity of figures has to be improved.

Response to the reviewer: Many thanks for the comments we have improved the graphs to be more clear.

Reviewer #6 wrote: The stability data of the sensor should be given in the Figures.

Response to the reviewer: Many thanks for the comments we have improved recent versions of the graphs contain the error bars.

Reviewer #6 wrote: What is the equivalent circuit of the EIS data? It should be given in the Figures.

Response to the reviewer: Many thanks for the comments, however, EIS experiment was not performed in research described in this manuscript.

Reviewer #6 wrote: The TEM or SEM images of Au/SAMmix/EDC-NHS-modified electrode should be given in the manuscript.

Response to the reviewer: Many thanks for the comments. Unfortunately, we do not have access to a sufficiently high-resolution TEM or SEM to be able to visualize clearly formed structures over Au/SAMmix/EDC-NHS-modified electrode.

Reviewer #6 wrote: In the section of Introduction, some related electrochemical works should be cited (such as Journal of The Electrochemical Society, 2022, 169(2): 027504; Journal of The Electrochemical Society, 2015, 162(7): B173-B179; Journal of Solid State Electrochemistry, 2014, 18(9): 2435-2442).

Response to the reviewer: Many thanks for the comments we have overviewed these and some other additional references recommended by reviewers.

Many thanks for the positive feedback.

We thank you for the attention you will pay to this revised version of the manuscript and we sincerely hope that our work after these revisions will be considered as relevant and attractive for publishing.

Yours sincerely,

Arunas Ramanavicius

----------------------------------------------------------------
Prof. habil. dr. Arunas Ramanavicius

Head of Department of Physical Chemistry,

Faculty of Chemistry and Geosciences, Vilnius University, 

e-mail: arunas.ramanavicius@chf.vu.lt

Reviewer 7 Report

Major comments:

1.       Describe the process of immobilization of BSA protein on the electrochemical sensor?

2.       How specific is the EDC-NHS linkage to SARS-CoV-2-rS protein?

3.       Describe how the linear range of ng/ul value in table 2 is compared to the nM value measured in the current study.

Minor comments:

1.       Line 79- please clarify the terminology “anti-rS”. Does it refer to the antibodies or protein?

2.       Figure 4: please mention the control and AntirS based on the color used (red or black)

Author Response

Manuscript ‘Electrochemical biosensor for the determination of specific antibodies against SARS-CoV-2 spike protein’

After recommended revisions

Response to reviewer #7:

We would like to thank the reviewer for the review of our manuscript, valuable comments and recommendations. We did our best in order to improve the manuscript according to minor revisions recommended by all three reviewers. All the most important changes are highlighted in the revised manuscript.

Reviewer #7 wrote: Describe the process of immobilization of BSA protein on the electrochemical sensor?

Response to the reviewer: Many thanks for the comments, we have improved, unused EDC-NHS groups were blocked by BSA protein.

Reviewer #7 wrote: How specific is the EDC-NHS linkage to SARS-CoV-2-rS protein?

Response to the reviewer: Many thanks for the comments we have improved EDC-NHS linkage is not specific to the SARS-CoV-2-rS protein, it will link to amine group of any protein.

Reviewer #7 wrote: Describe how the linear range of ng/ul value in table 2 is compared to the nM value measured in the current study.

Response to the reviewer: Many thanks for the comments, calculation of concentrations is presented.

Reviewer #7 wrote: Minor comments: Line 79- please clarify the terminology “anti-rS”. Does it refer to the antibodies or protein?

Response to the reviewer: Many thanks for the comments we have improved manuscript according yours recommendation.

Reviewer #7 wrote: Minor comments: Figure 4: please mention the control and AntirS based on the color used (red or black).

Response to the reviewer: Many thanks for the comments we have improved manuscript according yours recommendation.

Many thanks for the positive feedback.

We thank you for the attention you will pay to this revised version of the manuscript and we sincerely hope that our work after these revisions will be considered as relevant and attractive for publishing.

Yours sincerely,

Arunas Ramanavicius

----------------------------------------------------------------
Prof. habil. dr. Arunas Ramanavicius

Head of Department of Physical Chemistry,

Faculty of Chemistry and Geosciences, Vilnius University, 

e-mail: arunas.ramanavicius@chf.vu.lt

Round 2

Reviewer 3 Report

It can be accepted in the present version.

Reviewer 6 Report

All corrections have been made. This paper can be accepted.